# Use of Artificial Neural Networks to Optimize Stacking Sequence in UHMWPE Protections

**DOI:** 10.3390/polym13071012

**Published:** 2021-03-25

**Authors:** Jairo Peinado, Liu Jiao-Wang, Álvaro Olmedo, Carlos Santiuste

**Affiliations:** 1Department of Continuum Mechanics and Structural Analysis, Universidad Carlos III de Madrid, Avda. de la Universidad 30, Leganés, 28911 Madrid, Spain; jpeinado@fecsa.net (J.P.); lwang@pa.uc3m.es (L.J.-W.); 2FECSA Company Calle de Acacias 3, San Sebastián de los Reyes, 28703 Madrid, Spain; aolmedo@fecsa.net

**Keywords:** UHMWPE, impact, FEM, neural networks

## Abstract

The aim of the present work is to provide a methodology to evaluate the influence of stacking sequence on the ballistic performance of ultra-high molecular weight polyethylene (UHMWPE) protections. The proposed methodology is based on the combination of experimental tests, numerical modelling, and Artificial Neural Networks (ANN). High-velocity impact experimental tests were conducted to validate the numerical model. The validated Finite Element Method (FEM) model was used to provide data to train and to validate the ANN. Finally, the ANN was used to find the best stacking sequence combining layers of three UHMWPE materials with different qualities. The results showed that the three UHMWPE materials can be properly combined to provide a solution with a better ballistic performance than using only the material with highest quality. These results imply that costs can be reduced increasing the ballistic limit of the UHMWPE protections. When the weight ratios of the three materials remain constant, the optimal results occur when the highest-performance material is placed in the back face. Furthermore, ANN simulation showed that the optimal results occur when the weight ratio of the highest-performance material is 79.2%.

## 1. Introduction

In the global industry, ultra-high molecular weight polyethylene (UHMWPE) fibers are the leading high-performance fibers and widely used in armor applications. These fibers show great resistance against impact or blast loads being a promising alternative for aramids in armor products [1,2,3,4,5,6,7]. Czechowski et al. [8] compared aramid with UHMWPE (Dyneema) composites conducting a rigorously study including tensile, hardness, bending and delamination test. They determined that aramid has lower ultimate strain and higher value of hardness than UHMWPE. Thus, the behavior of UHMWPE composites under high velocity impact have received considerable attention in scientific literature [9,10].

Numerical models based on Finite Element Method (FEM) are an interesting approach to analyze the impact behavior of composites because these models help to understand the failure modes and energy absorption mechanisms [11,12,13]. The prediction of ballistic limit and back face signature under high velocity impacts are the main goals of FEM models when they are used to model UHMWPE protections [14,15,16]. However, the number of configurations that can be analyzed with FEM models is limited due to the computational cost.

When a composite panel is subjected to a high velocity impact, the role of front, middle and back layers are different. Therefore, design engineers use to combine different materials in the configuration of armor vests. UHMWPE vests, owed to its great flexibility and low density, give a good mobility and comfort to their users. The design of soft armors should take in consideration multi-threats like handgun calibers, shrapnel and/or protection against stabbing. Thus, finding the best combination of different UHMWPE plies to provide this kind of protection is the main concern of military industry [17]. There is a gap between industrial problems and scientific literature approaches in this topic because industry needs tools to find the best configuration under each threat, while academic researchers are studying the problem mainly with experimental and numerical methods that cannot be used to compare too many configurations.

Most of scientific works analyze the impact behavior of a single material, but only a few papers study the combination of different materials [18,19,20]. Aktaş et al. [18] analyzed six configurations of woven/knit fabric glass/epoxy hybrid composites. Results showed that specimens having outer layer of woven fabric exhibited the best impact performance. Randjbaran et al. [19] compared five configurations of hybrid composites combining Kevlar, carbon and glass layers. They found that stacking the first layer with glass fiber is better than to use Kevlar, the combination of carbon and glass is more efficient in central layers, and carbon fiber is not recommended at the last layer. Bandaru et al. [20] studied four configurations of hybrid composites combining Kevlar, glass and carbon fibers. They found that fixing the Kevlar layer at the rear side and carbon layer on the front side offers good ballistic impact resistance.

There are also some works that analyze hybrid composites including UHMWPE layers [21,22,23]. Zulkifli et al. [21] compared five hybrid composites combining carbon and HB26 UHMWPE layers. They found that adding a few layers reinforced with carbon fibers in front face results in a significant 30% reduction in back-face signature with a more than two times improvement in flexural yield strength.

Yang and Chen [22] studied two armor hybrid packages manufactured with aramid fabrics and UHMWPE unidirectional layers. In their work, a decrease of 31% in the back-face signature can be reached, on the same areal density, when aramid layers are placed in the front position and UHMWPE on rear.

Chen et al. [23] studied two hybrid UHMWPE panels combining woven and unidirectional layers. The main conclusion is that hybridization can enhance the ballistic results. The optimum ratio of woven to unidirectional layers is 1:3. The best configuration is using woven layers in the front face and unidirectional layers in the rear face because woven structures highlight better shear strength and unidirectional structures show better tensile strength.

However, these studies are based on experimental techniques or FEM models, thus the number of combinations is limited to a maximum of six cases. The use of Artificial Neural Networks (ANN) is a good approach to analyze the impact behavior of numerous configurations of composite protections. This technique has been successfully used for quasi-static and dynamic problems as in Birecikli et al. [24] for predicting the ultimate failure load on laminate stacking sequence of glass fiber composite on bonding joints exposed to tensile strength; Naderpour et al. [25] where ANN were used to predict the compressive strength of FRP-confined concrete samples; Malik & Arif [26] used ANN to analyze low-velocity impacts; Bezerra et al. [27] studied shear mechanical properties with ANN; and Galatas et al. [28] used ANN to accurately predict the mechanical properties of sandwich-structured composite

Moreover, ANN have been used in the analysis of carbon fiber reinforced plastics (CFRPs) under high velocity impact [29,30]. Fernández-Fernández et al. [29] use ANN to predict the ballistic limit of CFRPs as a function of impact angle, and Artero-Guerrero et al. [30] studied 2048 cases to analyze the influence of stacking sequence on the ballistic limit of CFRPs. They used only one material and the objective was to determine the ballistic limit as a function of the orientation of each layer.

The present work proposes a methodology similar to that used by Artero-Guerrero et al. [30] which consists on the combination of ANN, FEM modelling and experimental tests. The main contribution of the present work is the consideration of different materials and different areal densities which implies the use of different input variables. The experimental tests are conducted to validate the FEM model. The validated FEM model is used to provide data for the ANN, some data provided by FEM model are used to train the ANN and other data are used to validate the ANN and to check its accuracy.

The types of UHMWPE layers are combined in 361 configurations to study the influence of the stacking sequence on the ballistic limit. The effect of the position of each layer, the weight ratio of each material, and the areal density are studied. Conclusions can be used by design engineers to improve ballistic performance of UHMWPE protections.

## 2. Methodology

The methodology used in this work combines three approaches: experimental (high-velocity impact tests), numerical (modelling based on Finite Element Method), and machine learning (Artificial Neural Network). The large amount of data required to train and validate the ANN is unaffordable from an experimental point of view because the objective is the prediction of ballistic limit as a function of stacking sequence and the estimation of ballistic limit using experimental methods requires at least seven tests for each configuration. For this reason, a numerical model is needed to provide enough number of results for the training of the ANN. To increase the reliability of the results, experimental tests are conducted to validate the numerical model. This combination allows to predict the ballistic limit of numerous UHMWPE configurations with high accuracy and low cost.

In this work, three UHMWPE materials (PE1, PE2, and PE3) are combined to analyze the influence of stacking sequence, and the mechanical properties are shown in Table 1. There are unlimited combinations if the areal density and the position of the materials are modified, thus, to limit the number of configurations two studies are conducted:**Influence of layers position**. In the first study, the areal density and the number of layers of each material were fixed to analyze the position of each material. A total of 252 configurations were studied by combining two layers of PE1, one layer of PE2 and six layers of PE3.**Influence of weight ratios**. In the second study, the areal density and the weight ratio of each material are analyzed. A total of 109 configurations were studied by combining PE1, PE2 and PE3. In this study, all the layers of the same material were packaged in a single block.

### 2.1. Experimental Procedure

High-velocity impact tests on UHMWPE panels were carried out to validate the FEM model. A gas gun was used to accelerate a 7.5 mm diameter spherical steel projectile with a mass of 1.73 g. The helium pressure was modified to have a wide range of impact velocities from 350 m/s to 810 m/s. The specimen dimension was 129 × 129 mm^2^ and specimens were clamped into a steel frame with a free area of 100 × 100 mm^2^. Two high speed video cameras (Photron Ultima APX-RS at 50 kfps, Photron Europe Limited, UK) were used to measure impact and residual velocities.

Three different UHMWPE materials used in the design of soft armor vests were selected by FECSA, which is a company with wide experience providing ballistic protections to police and armed forces, they selected and delivered the materials for experimental tests. The commercial designation of UHMWPE materials has been omitted due to confidentiality. Thus, they are named PE1, PE2 and PE3 in this work. Material properties are shown in Table 1.

These materials were combined in six stacking sequences: ([3·PE3/8·PE1], [7·PE3/3·PE1], [3·PE1/7·PE3], [1·PE2/8·PE3], [6·PE3/2·PE1/1·PE2], and [2·PE1/1·PE2/6·PE3]). The areal densities varied from 1808 g/m^2^ to 1981 g/m^2^. The six stacking sequences include the combination of two types of UHMWPE (PE1 with PE3, and PE2 with PE3) and the combination of three types of UHMWPE (PE1, PE2 and PE3). The purpose was to consider the possibility of selection of different materials. They also include combinations of same materials but with different weight ratios and stacking sequences. The numerical model can be considered validated if it is able to accurately predict the ballistic limit of these six stacking sequences.

### 2.2. Numerical Model

A FEM model was developed in ABAQUS/Explicit considering two solids: steel projectile and UHMWPE laminate. The spherical projectile was modeled as a linear elastic material (ρ = 7850 kg/m^3^; E = 210 GPa and υ = 0.3) because plastic deformation was not observed in the experimental tests. Only the free area of the clamp was modelled for simplification (100 × 100 mm^2^) and clamped boundary conditions were applied to the four edges. Dynamic explicit analysis was carried out including geometric non-linearity and large deformations options. The spherical projectile was meshed using linear tetrahedral elements (C3D4 in ABAQUS/Explicit notation) while UHMWPE laminate was meshed with 8-node brick elements with reduced integration (C3D8R in ABAQUS/Explicit notation). A mesh sensibility analysis was conducted to find a compromise between accuracy and computational cost. The finest mesh with more than 60,000 elements per layer provided accurate results, however the computational cost was about 30 h in a workstation with 24 CPUs. The element size was kept constant in the impact area, but the element size was increased in the boundaries of the plates to look for a more efficient mesh. The selected mesh consisted in a total of 16,838 elements per layer with a non-structured distribution and a finer mesh in the contact area that implied a computational cost of 6 h, see Figure 1.

Surface-node surface contact algorithm available in ABAQUS/Explicit was used to model the interaction between projectile and UHMWPE laminate. In addition, a self-contact condition was used to avoid penetration between eroded UHMWPE elements. Moreover, a contact between layers of UHMWPE was considered in the model with a friction coefficient equal to 0.175 [31].

The anisotropic UHMWPE layers were modelled as elastic behavior up to failure. Failure was predicted with a modification of Hou failure criteria [32] including four failure modes: fiber failure, matrix cracking, matrix crushing and delamination, assuming a quadratic interaction between stresses in the four failure modes. Hou failure criteria was implemented in a VUMAT subroutine developed in ABAQUS/Explicit including also a procedure to degrade material properties when any of the failure criterion was verified. The mechanical properties subjected to the degradation procedure depends on the failure criterion that is verified. For instance, fiber failure produces the degradation of all the properties while delamination only degrades out-of-plane properties.

The simulation of ballistic impacts, which can imply the perforation of the target, requires also the use of an erosion criterion. When damage occurs in a finite element, the stresses drop to values close to zero and large deformations appear. These damaged elements do not contribute to reduce the kinetic energy of the projectile, but they can cause lack of convergence during simulation and instability problems, thus a maximum strain criterion was adopted to remove the distorted elements. A similar procedure was previously implemented in the modelling of impacts on composites reinforced with carbon fibers [11] and glass fibers [12]. The modelling of composites under dynamic loads can also include thermal softening [33] and the influence of strain rate [34], but these effects were not included in the model to reduce computational cost because numerous configurations were needed to train the ANN. It should be noticed that thermal softening and influence of strain rate are needed to obtain accurate results in terms of damage extension, but the ballistic limit can be accurately predicted using steel spherical projectiles if these effects are neglected.

### 2.3. Artificial Neural Network

Within this work, an Artificial Neural Network was used to predict the ballistic limit of UHMWPE laminates with different stacking sequences. An ANN is a machine learning algorithm that process the input information to produce an answer. The validity of the answer depends on how the information has been treated; this process is called training or learning. In this training process, as in the human brain, the internal structure is modified to assure the correct answer learning by examples. The learning capability is reached, also as in the human brain, by means of a highly interconnected internal structure. The ability of ANN model to learn by highly non-linear examples, noisy data, or fitting problems such as what could be found in solid mechanics represents a good approach to solve impact problems.

The ANN architecture selected in this work is a Multi-Layer Perceptron in combination with the Feed-Forward Back Propagation algorithm [35]. A set of interconnected neurons (perceptrons) receives signals from previous layers of neurons or external input, which are interpreted, processed, and transmitted to the adjacent layers, or to final output cell.

A neuron is a real function of the input vector (*y1*, *y2*, …, *yk*) that can be described as *Z_i_* = *f_i_*(∑*ω_ij_y_j_* − *b_i_*), where *ω_ij_* shall be the weight which is dependent on the link from the previous adjacent neurons, and *b_i_* shall be the threshold that needs to be surpassed by the sum of the weighed inputs to activate the neuron output *Z_i_* by means of the activation function *f_i_*.

Their structure is modified during the training process to minimize the output Mean Square Error,
(1)$$MSE=\;\frac{1}{N}\sum {\left({\hat{z}}_{i}-z_{i}\right)}^{2},$$
where *z_i_* represents the target output, and ${\hat{z}}_{i}$ represent the output array.

The activation functions are tangent sigmoid for input and hidden layers. These activation functions were selected because they are widely used and universally proven approximators [17,26,34]:(2)$$f\left(x\right)=\;\frac{2}{\left(1+{℮}^{-2x}\right)}-1,$$

And linear function (pureline) for output cell:(3)f(x)=x,

Levenverg-Marquardt iterative algorithm is deployed to minimize the error between the output responses and target outputs. During the iterations, a reduce set of the global input $A=\;\left\{{\left(x,y\right)}_{r}^{A}\;|\;r=1,\ldots ,k\right\}\subset S$ is split into two randomly subsets for learning $B=\;\left\{{\left(x,y\right)}_{r}^{B}\;|\;r=1,\ldots ,l\right\}\subset A$ and cross-validation $C=\;\left\{{\left(x,y\right)}_{r}^{C}\;|\;r=1,\ldots ,m\right\}\subset A$, so the first is used to calculate the weights and thresholds, and the second is used to calculate the Mean Square Error for the predictions of the reduce set A:

If g(r) is the function containing the individual error terms, known at point r, then E(r) is the objective error function for the outputs:(4)$$E\left(r\right)=\;\sum {\left({\hat{z}}_{i}-z_{i}\right)}^{2}={\left\|g\left(r\right)\right\|}^{2},$$

And the values for new weights and thresholds are:(5)$$\omega_{ij}\left(r+1\right)=\;\omega_{ij}\left(r\right)-\lambda \frac{\partial g\left(r\right)}{\partial \omega_{ij}}$$
(6)$$b_{i}\left(r+1\right)=\;b_{i}\left(r\right)-\lambda \frac{\partial g\left(r\right)}{\partial b_{i}}$$
where λ is the scalar Marquardt parameter that is decreased after each epoch to update both, weights and thresholds in the direction in which the performance function decreases most (gradient descend) [36].

These steps are repeated computing the $MSE^{B}$ consecutively in each epoch until a stable value is reached.

In this work, two MLP have been developed with MATLAB Neural Network Toolbox to predict the ballistic limit of the stacking sequences corresponding to the two studies: influence of layers position and influence of weight ratios.

In the influence of layers position study, all the stacking sequences had 9 layers, thus 9 input variables were considered on the first MLP. The value of these variables can be 1 if the corresponding layer was of PE1, 2 for PE2, and 3 for PE3.

In the influence of weight ratios study, four input variables were considered. The first three variables were the areal density of PE1, PE2 and PE3 materials, while the fourth variable corresponds to the relative position of the materials, see Table 2.

The training process was carried out following three steps. In the first step, the size of the training set was stablished, the first training sets were 40 stacking sequences of 252 for the first study and 16 stacking sequences of 109 for the second study. The FEM model was used to obtain the ballistic limit of 46 cases in the first study and 23 in the second study because 6 and 7 cases were used for validation in the first and second studies respectively. In the second step, the training sets were provided to ANN. Then, MATLAB Tool separates internally the provided data into three subsets: 70% of data are used to train the ANN during the current epoch, another 15% of data are used for testing, which stay untrained until the next epoch is reached, and 15% of data are used for cross-validation that compares the results of these subsets with the prediction made by the ANN. This iterative process was repeated for 1000 epochs, but this does not guarantee the stabilization of the correct solution, thus this process must be repeated several runs until and accurate solution is reached. In this work, the number of runs was determined by a coefficient of determination (R^2^) higher than 99%. The third step consisted in the validation of the ANN comparing the predictions of the ANN with the results of the FEM model. If the ANN predictions agreed with FEM model results, then the ANN was considered validated. On the other hand, if the ANN was not validated the size of the training set was increased and the three steps were repeated until the ANN predictions agreed with FEM results. The final size of the training sets were 50 and 25 stacking sequences for the first and second study respectively.

The MLP architecture used in this paper consists of three layers of neurons: input, hidden and output layers, see Figure 2. The number of input neurons is equal to the number of input variables (9 in the study of layers position and 4 in the study of weight ratios) and the number of output neurons is equal to the number of output variables, in this work only one output variable is used: ballistic limit. The number of hidden neurons has been chosen after an optimization process in which, with an initial number of $N_{h}=\sqrt{NM}$ where N and M are the number of neurons in the input and output layers respectively [37], being $N_{h1}=3$ for the first MLP and $N_{h2}=2$ for the second. Nevertheless, this value is iteratively modified minimizing the computed global Root Mean Square Error, calculated as $RMSE=\sqrt{MSE}$. The valuation ended when the stop criterion is reached, in this case, when a global error is below 0.5% for the first approach and 2% for the second. The number of hidden neurons was established for each training set, when the training set was increased the number of hidden neurons had to be recalculated. For the final training sets, 10 hidden neurons were used for the first MLP and 14 hidden neurons for the second study. Finally, a value of 1000 epochs was selected for both MLPs, λ=0.4 and a $MSE^{B}=10^{-7}$ were assumed accordingly with [27].

The input and output vectors in the first study have the size $y_{ij}={\left[Y\right]}_{9xn}$, and $z_{i}={\left[Z\right]}_{1xn}$ because there are nine input variables (the material of each of the nine layers) and the input and output vectors in the second study are $y_{ij}={\left[Y\right]}_{4xn}$, and $z_{i}={\left[Z\right]}_{1xn}$ because there are four input variables (PE1 areal density, PE2 areal density, PE3 areal density, relative position of each material).

Nguyen-Widrow approach was used for weights and thresholds initialization, in which values between −0.5 and 0.5 or −1 and 1 are chosen randomly depending on the scalar parameter β, which follows the equation β=0.7·p1n, where *p* and *n* are the number of variables in the hidden layer, and the number of input variables respectively.

## 3. Validation

This section includes experimental, numerical, and ANN results obtained in the prediction of the ballistic limit of UHMWPE panels with different stacking sequences combining PE1, PE2 and PE3 layers. The results are divided in three sections: first, the validation of the FEM numerical model by the comparison of numerical predictions and experimental results; second, the ANN training process using a set of results provided by the FEM numerical model; and finally, the validation of the ANN predictions using other results obtained with the numerical model.

### 3.1. Validation of FEM Model

To validate the numerical model, each stacking sequence subjected to experimental tests, as depicted in Table 3, was simulated under different impact velocities. The residual velocities predicted by the numerical model were compared with experimental results. The residual velocity is the key parameter to validate the model because it represents the energy dissipated by the laminate through its failure mechanisms. The numerical ballistic limit was established as the maximum impact velocity that does not produce the projectile penetration. The experimental ballistic limit was obtained from the residual velocity using the Lambert-Jonas equation [38]:(7)vres= a·(vres1p−vbl1p)p,
where a, *p* and *v_bl_* are fitting parameters, being *v_bl_* the experimental ballistic limit.

Figure 3 shows numerical and experimental results of the residual velocity as a function of the impact velocity for the stacking sequence [2·PE1/1·PE2/6·PE3]. An excellent agreement between numerical model predictions and experimental results can be observed. Residual velocity is zero for impact velocities lower than the ballistic limit, while once the ballistic limit is reached, a sharp slope is observed in the residual velocity. Finally, as impact velocity increases, the residual velocity tends to a constant slope. An accurate prediction of the residual velocity is required for a good prediction of ballistic limit. The difference between experimental and numerical ballistic limits is 2.3% for this stacking sequence.

Table 3 shows experimental and numerical ballistic limits for all the stacking sequences. Maximum error is only 3.6%, thus ballistic limit is accurately predicted in all the analyzed cases. Moreover, the same trends can be observed in numerical and experimental results. When PE3 is in the back face, the ballistic limit is higher than when PE3 is in the front face, this effect can be seen comparing [3·PE1/7·PE3] with [7·PE3/3·PE1] and comparing [6·PE3/2·PE1/1·PE2] with [2·PE1/1·PE2/6·PE3]. Ballistic limit also increases with PE3 weight ratio and this effect can be observed comparing [3·PE3/8·PE1] with [7·PE3/3·PE1].

### 3.2. Validation of ANN

Two virtual testing campaigns were carried out with the validated FEM model to provide the data required for the training process of the MLPs. For the first study, influence of layers position, the results of 50 stacking sequences were used to train the MLP. In the second study, influence of weight ratios, the MLP was trained using the results of 25 stacking sequences. For each stacking sequences, at least a set of 7 simulations with different impact velocities were conducted to calculate the ballistic limit used to train the MLPs.

The results of the training process in the first study, influence of layers position, are shown in Figure 4 where ballistic limit predicted by ANN is plotted as a function of the ballistic limit obtained by the FEM model. The results include the 50 cases used in the training process plus 6 more cases used to validate the ANN (the results of these 6 additional cases were not provided to the MLP during training process). All the cases are inside the marked region that represents a maximum error of 1%, being the maximum error 0.89%. The average error of the 6 cases used for validation is 0.55%. Hence, it can be concluded that the ANN provides an accurate prediction of the ballistic limit and it can be used to analyze the results of different stacking sequences to find the best configurations.

Figure 5 shows the results of the training process in the second study, influence of weight ratios. The results include the 25 cases used in the training process plus 7 more cases used to validate the ANN (the results of these 7 additional cases were not provided to the MLP during training process). In this study, the maximum error is 2.37% and the average error in these 7 cases used of validation is 1.36%. The errors in this study are higher because the areal density and the number of layers of each material are not constant, thus it is a more complex problem. Nevertheless, all the results predicted by the ANN are in excellent agreement with the FEM model results.

## 4. Results and Discussion

### 4.1. Influence of Layers Position

Figure 6 shows the results of ballistic limit predicted by the ANN for all the stacking sequences composed by 2 layers of PE1, one layer of PE2, and 6 layers of PE3. The 252 stacking sequences are divided into 9 colors, each one corresponds to the position of the PE2 layer. There is a clear trend inside each color: the ballistic limit decreases with laminate number. The order of the stacking sequences has a clear influence on this trend because in the first cases, all the PE3 layers are in the back face, while in the last cases the PE3 layers are in the front face. The maximum ballistic limit is 513.1 m/s and the minimum 463.3 m/s, thus differences of 10.7% can be found in laminates with the same materials and the proper stacking sequence can significantly improve the ballistic performance of the UHMWPE panel.

To get a better understanding of this phenomenon, all the stacking sequences were reordered as a function of the position of PE3 layers. Figure 7 shows the ballistic limit predicted by the ANN as a function of the position of PE3 layers, the first cases correspond to stacking sequences with PE3 layers at the back face, while the last cases correspond to stacking sequences with PE3 layers at the front face. There are only 13 stacking sequences with a ballistic limit higher than 505 m/s and all of them are in the left half of Figure 7. On the other hand, there are only 17 stacking sequences with a ballistic limit lower than 480 m/s and all of them are in the right half of Figure 7. Thus, it can be concluded that the position of PE3 layers has a clear influence on ballistic limit, PE3 layers must be located at the back face.

A new variable was defined to quantify the influence of the position of each layer. The energy absorbed capacity of each layer is proportional to the tensile strength and the ultimate strain, thus the absorbed energy variable (AEV) for each layer has been defined as follows:(8)$${EAV}_{i}=X_{T,i}\cdot \varepsilon_{u,i}\cdot p_{i},$$
where ***X_T_*** is the tensile strength, ***ε_u_*** is the ultimate strain, and ***p*** is a new proposed parameter whose value is equal to the position of the layer divided by 10. The objective of this variable is to consider that the layers in the back face have a higher contribution to the ballistic limit than layers in the front face. Considering the contribution of the nine layers, the absorbed energy variable for each stacking sequence can be calculated according to Equation (9):(9)$$EAV=\sum_{i=1}^{9}X_{T,i}\cdot \varepsilon_{u,i}\cdot p_{i},$$

Figure 8 shows the ballistic limit predicted by ANN as a function of the absorbed energy variable. There is a clear tendency that can be observed, the ballistic limit increases with the absorbed energy variable. Thus, according to the results, the materials with a higher energy absorption capability must be used in the back face of the laminate.

Table 4 shows the results of the 10 best stacking sequences of this study in black. All of them have PE3 in the back face and six of them have PE1 in the front face. The best three stacking sequences have all the PE3 in the back face, and the other seven stacking sequences have five of the six PE3 layers at back face. Thus, the key to optimize the stacking sequence is the use of PE3 in the back face.

### 4.2. Influence of Weight Ratios

The number of configurations considering different weight ratios of PE1, PE2 and PE3 layers is unlimited. To have a better understanding of the influence of the weight ratio of each material is not necessary to analyze all the combinations, thus, only configurations with an areal density around 1850 g/m^2^ were studied. The minimum areal density was 1768 g/m^2^ and the maximum was 1987 g/m^2^. Moreover, in all the studied configurations, the layers of the same material were stacked together. In this study, stacking sequences with different areal densities are compared, thus the variable used in the analysis is the specific ballistic limit, defined as the ballistic limit divided by the areal density.

Figure 9 shows the results of specific ballistic limit predicted by the ANN for all the stacking sequences considered in the study of the influence of weight ratios. The maximum specific ballistic limit is 0.325 m^3^/g·s and the minimum 0.196 m^3^/g·s, the difference between minimum and maximum is 65.9%. Therefore, the proper selection of materials weight ratios can improve the ballistic performance of UHMWPE protections in more than 60%. The 109 stacking sequences are divided into 4 groups. A configuration of pure PE3 was included in this study because this material has the best impact properties, and it is also the most expensive. Thus, one of the goals of this study is searching cheaper configurations with best impact behavior. The results of the pure PE3 panel with an areal density of 1944 g/m^2^ are ballistic limit of 520 m/s and specific ballistic limit of 0.267 m^3^/g·s (this case was used in the training process). It should be notice, that the pure PE3 panel is the most expensive but there are numerous configurations with better ballistic performance.

There are 16 combinations of PE1 and PE3, with PE3 weight ratios from 11.9% to 92.3%, two of them with a specific ballistic limit higher than pure PE3. Therefore, the ballistic performance of pure PE3 can be improved reducing the cost, however, these two combinations, [8·PE3/PE1] and [PE1/8·PE3], have a high PE3 weight ratio: 92.3%. Figure 10 shows the specific ballistic limit of stacking sequences combining PE1 and PE3 as a function of PE3 weight ratio, it can be observed that specific ballistic limit increases with PE3 weight ratio. The optimum weight ratio of PE3 is around 90% because the results with 92.3% of PE3 are better than pure PE3. Moreover, the results with PE3 in front face or back face are quite similar, thus the influence of PE3 weight ratio is higher than the influence of PE3 position.

14 combinations of PE2 and PE3 are included in the study, four of them with higher specific ballistic limit than pure PE3 and PE3 weight ratios between 71.9% and 87.2%. Thus, the PE3 weight ratio can be more reduced if it is combined with PE2 than with PE1. Figure 11 shows the specific ballistic limit of the stacking sequences that combine PE2 and PE3 as a function of PE3 weight ratio. Specific ballistic limit increases with PE3 weight ratio until 70%, then there a plateau between 70% and 90%. These four stacking sequences with PE3 weight ratio between 70% and 90% are those with better ballistic performance than pure PE3. Again, the influence of PE3 position is negligible comparing with the influence of weight ratios.

There are also 78 configurations that combine PE1, PE2, and PE3, 25 of them with higher specific ballistic limit than pure PE3, and PE3 weight ratios between 35.9% and 79.2%. These results indicate that the best option to improve the results of PE3 is the combination of the three materials. Figure 12 shows the results of stacking sequences combining the three materials as a function of PE3 weight ratio. There is not a clear trend, the specific ballistic limit increases with PE3 weight ratio from 10% to 35%, then specific ballistic limit decreases until 60% and it increases again up to a maximum value at 80% approximately. The conclusion is that there are different combinations of PE1, PE2, and PE3, that can lead to a better performance than pure PE3 with weight ratios between 35% and 80%, however, most of the stacking sequences with higher specific ballistic limit than pure PE3 have weight ratios between 65% and 80%. Therefore, the best option is the combination of the three materials but there is not a clear rule to combine them.

Table 4 shows in blue the stacking sequences with the best ballistic performance in terms of specific ballistic limit. Five of them, including the 3 best results, have a PE3 weight ratio of 79.2%. Three stacking sequences have a PE3 weight ratio of 66.6% and two laminates have a PE3 weight ratio of 33.3%. The influence of the PE3 position is not clear because in four cases it is positioned in front face, in four cases it is in the middle and in two cases it is in the back face. All of the 10 stacking sequences with highest specific ballistic limit combine the three materials with a PE1 weight ratio lower than 10%. Therefore, the best configurations use PE1 but with a low weight ratio.

## 5. Conclusions

A methodology that combines experimental tests, FEM modelling and ANN simulation, is validated to evaluate the ballistic performance of 361 stacking sequences formed by three different UHMWPE materials. The main conclusions drawn from the analysis of the results are as follows:The FEM model can be considered validated because it is able to reproduce the influence of stacking sequence on ballistic limit.The ANN developed using a combined methodology predicts the ballistic limit of different stacking sequences successfully.The trained ANN shows a great capability to analysis a huge number of stacking sequences. In the analysis of 252 stacking sequences combining 3 UHMWPE materials with the same weight ratio, it has been observed that ballistic performance can be improved by 10.7% without any weight increase.When the weight ratio of the materials is constant, placing the PE3 layers on the back face is the most effective configuration.The analysis of 109 stacking sequences with different weight ratios revealed that when the three materials are properly combined the ballistic performance is better than using only the material with highest quality. Thus, ANN simulations can be used to increase the ballistic limit with lower cost.

## Figures and Tables

**Figure 1 polymers-13-01012-f001:**
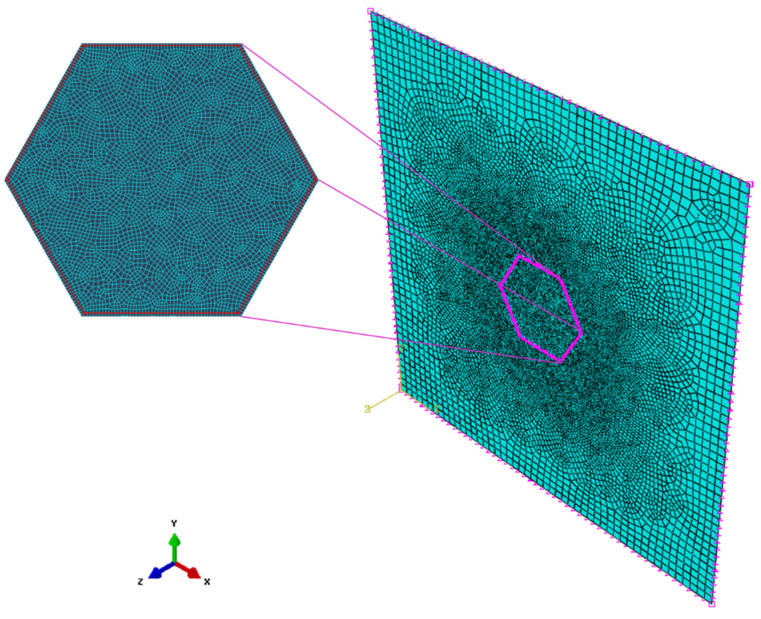
Mesh of a single layer the numerical model with a total of 16,838 8-nodes brick elements (C3D8R) per layer.

**Figure 2 polymers-13-01012-f002:**
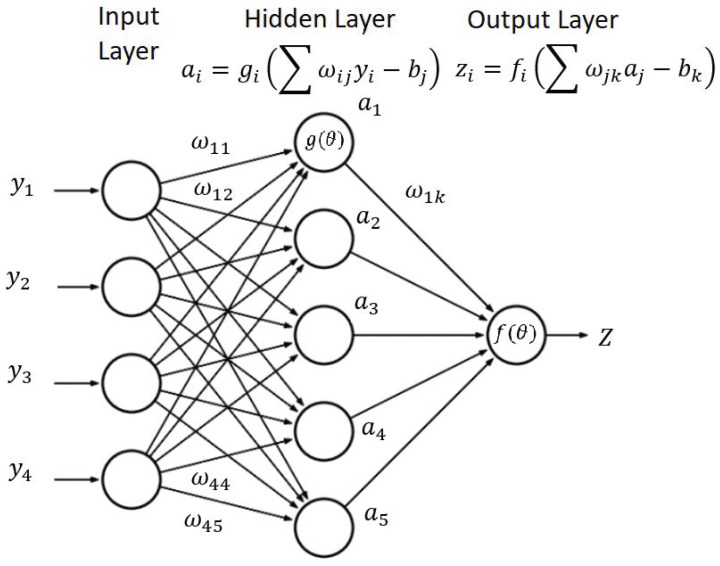
Scheme of the MLP used to predict the residual velocity of the projectile.

**Figure 3 polymers-13-01012-f003:**
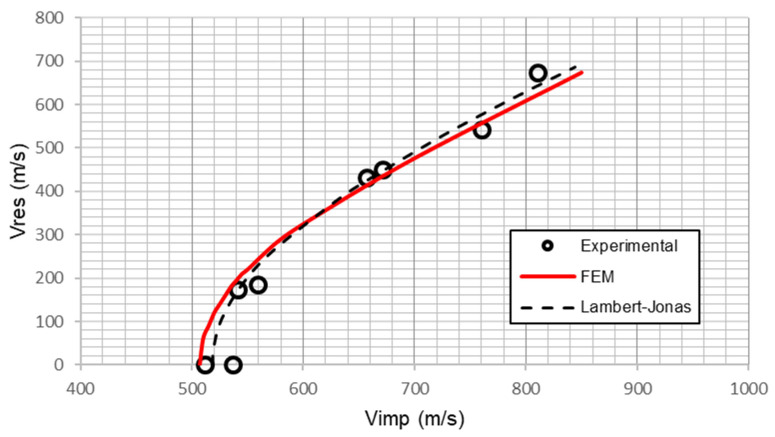
Residual velocity versus impact velocity for the stacking sequence [2·PE1/1·PE2/6·PE3]. Experimental results and numerical predictions.

**Figure 4 polymers-13-01012-f004:**
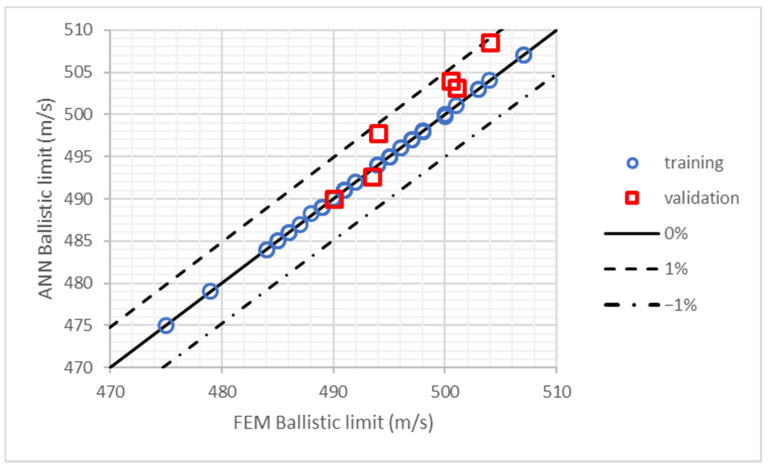
Training and validation of the ANN used in the study of layers position. Ballistic limits predicted by FEM model and ANN.

**Figure 5 polymers-13-01012-f005:**
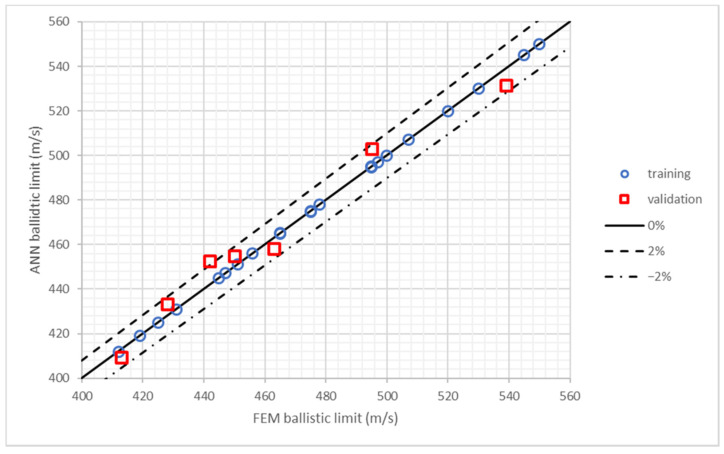
Training and validation of the ANN used in the study of weight ratios. Ballistic limits predicted by FEM model and ANN.

**Figure 6 polymers-13-01012-f006:**
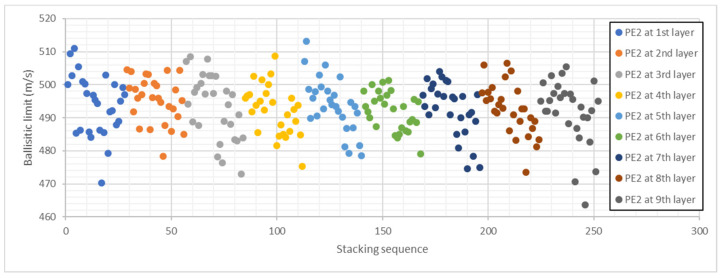
Ballistic limit predicted by ANN for all the stacking sequences with two layers of PE1, one layer of PE2, and six layers of PE3. Stacking sequences ordered as a function of PE2 layer position.

**Figure 7 polymers-13-01012-f007:**
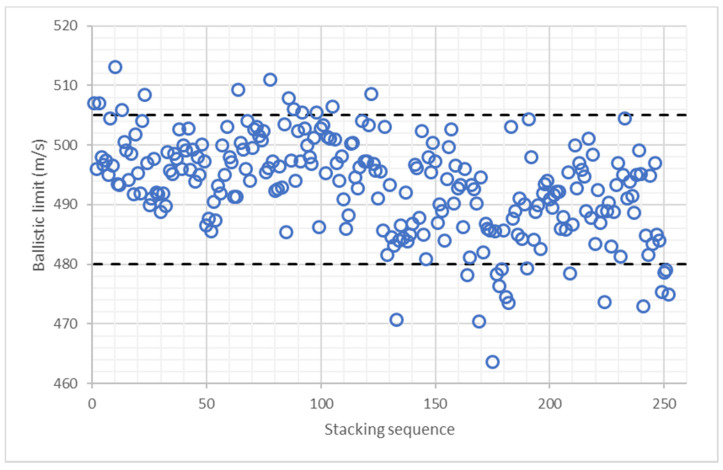
Ballistic limit predicted by ANN for all the stacking sequences with 2 layers of PE1, one layer of PE2, and 6 layers of PE3. Stacking sequences ordered as a function of PE3 layers position.

**Figure 8 polymers-13-01012-f008:**
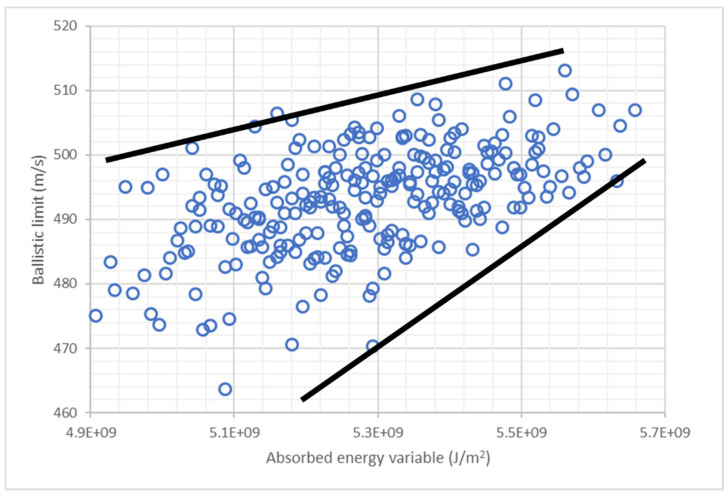
Ballistic limit predicted by ANN for all the stacking sequences with 2 layers of PE1, one layer of PE2, and 6 layers of PE3. Influence of absorbed energy variable.

**Figure 9 polymers-13-01012-f009:**
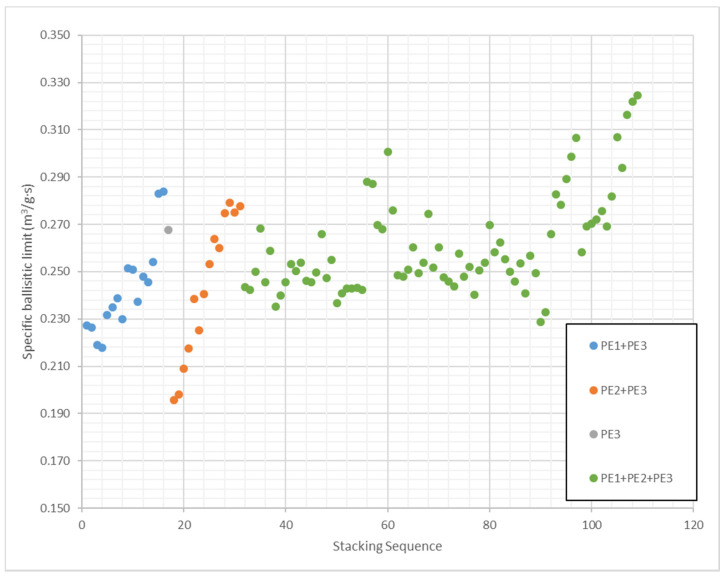
Specific ballistic limit predicted by ANN for stacking sequences considering different weight ratios of PE1, PE2 and PE3.

**Figure 10 polymers-13-01012-f010:**
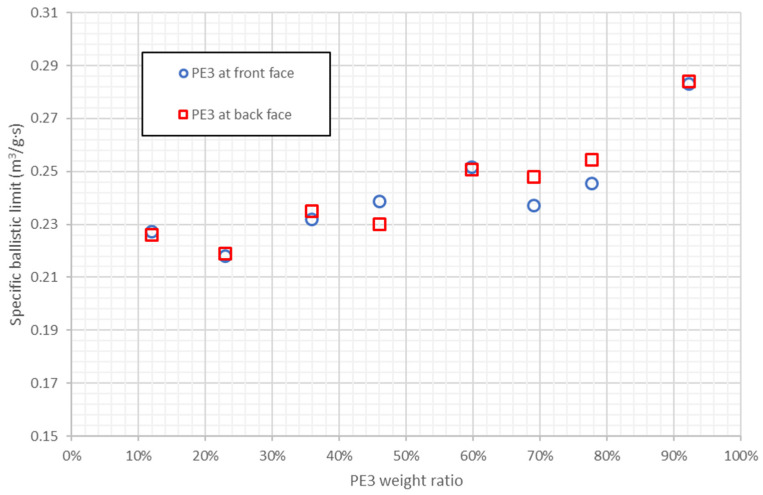
Specific ballistic limit predicted by ANN for stacking sequences combining PE1 and PE3. Results as a function of PE3 weight ratio.

**Figure 11 polymers-13-01012-f011:**
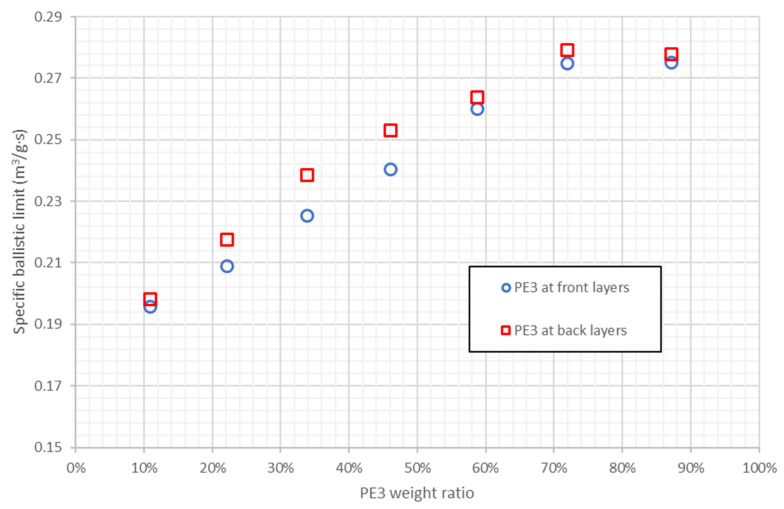
Specific ballistic limit predicted by ANN for stacking sequences combining PE2 and PE3. Results as a function of PE3 weight ratio.

**Figure 12 polymers-13-01012-f012:**
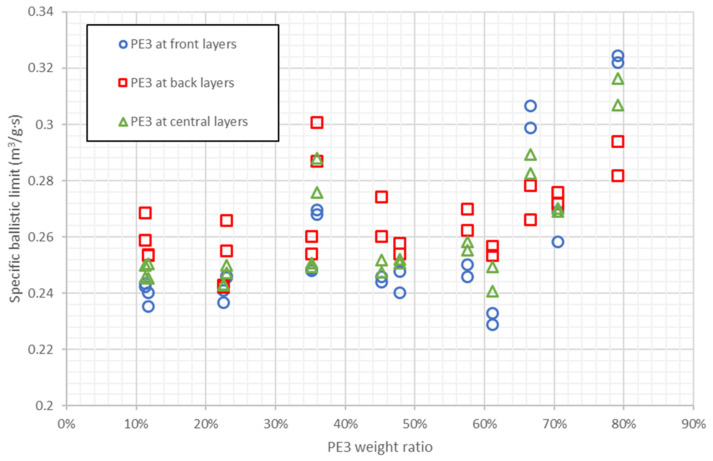
Specific ballistic limit predicted by ANN for stacking sequences combining PE1, PE2 and PE3. Results as a function of PE3 weight ratio.

**Table 1 polymers-13-01012-t001:** Material properties of UHMWPE materials.

	PE1	PE2	PE3
Areal density (g/m^2^)	253	145.16	216.52
Thickness (mm)	0.3	0.18	0.252
E11 (GPa)	78.997	52.591	55.185
Xt (MPa)	2807.1	1498.8	1631.8
Ultimate strain (−)	0.3	0.7	0.8

**Table 2 polymers-13-01012-t002:** Values of the fourth input variable as a function of the relative position of the three UHMWPE materials.

4th Variable	Front Face	Middle	Back Face
1	PE3	PE2	PE1
2	PE3	PE1	PE2
3	PE2	PE3	PE1
4	PE1	PE3	PE2
5	PE2	PE1	PE3
6	PE1	PE2	PE3
7	PE3	0	PE2
8	PE3	0	PE1
9	PE2	0	PE3
10	PE1	0	PE3
11	PE3	0	0

**Table 3 polymers-13-01012-t003:** Stacking sequences of UHMWPE impacted in the experimental tests. Numerical and experimental results.

	Areal Density	Weight Ratio (%)			Ballistic Limit (m/s)		
Stacking Sequence	(g/m^2^)	PE1	PE2	PE3	FEM	Exp.	Error
3·PE1/7·PE3	1947	22.34	0	77.66	495	483.1	−2.4%
7·PE3/3·PE1	1947	22.34	0	77.66	478	478.8	0.2%
3·PE3/8·PE1	1808	64.16	0	35.84	419	434.1	3.6%
1·PE2/8·PE3	1981	0	12.77	87.23	550	567.3	3.1%
2·PE1/1·PE2/6·PE3	1839	15.77	13.76	70.47	507	518.5	2.3%
6·PE3/2·PE1/1·PE2	1839	15.77	13.76	70.47	495	489.6	−1.1%

**Table 4 polymers-13-01012-t004:** Stacking sequences of UHMWPE impacted in the experimental tests. Numerical and experimental results.

	Weight Ratio			Areal Density	Ballistic Limit	Specific Ballistic Limit
Stacking Sequence	PE1	PE2	PE3	(g/m^3^)	(m/s)	(m^3^/g·s)
[2·PE1/PE2/6·PE3]	15.8%	13.8%	70.5%	1839	513.1	0.279
[PE1/PE2/PE1/6·PE3]	15.8%	13.8%	70.5%	1839	511.0	0.278
[PE2/2·PE1/6·PE3]	15.8%	13.8%	70.5%	1839	509.4	0.277
[2·PE1/PE3/PE2/5·PE3]	15.8%	13.8%	70.5%	1839	508.6	0.277
[PE1/PE3/PE1/PE2/5·PE3]	15.8%	13.8%	70.5%	1839	508.5	0.276
[PE3/2·PE1/PE2/5·PE3]	15.8%	13.8%	70.5%	1839	507.8	0.276
[PE2/PE1/PE3/PE1/5·PE3]	15.8%	13.8%	70.5%	1839	507.0	0.276
[PE1/PE2/PE3/PE1/5·PE3]	15.8%	13.8%	70.5%	1839	507.0	0.276
[PE1/PE3/PE2/PE1/5·PE3]	15.8%	13.8%	70.5%	1839	506.5	0.275
[PE3/PE1/PE2/PE1/5·PE3]	15.8%	13.8%	70.5%	1839	506.1	0.275
[7·PE3/1·PE2/1·PE1]	7.6%	13.2%	79.2%	1910	619.9	0.325
[7·PE3/1·PE1/1·PE2]	7.6%	13.2%	79.2%	1910	614.9	0.322
[1·PE2/7·PE3/1·PE1]	7.6%	13.2%	79.2%	1910	604.2	0.316
[6·PE3/2·PE2/1·PE1]	7.4%	26.0%	66.6%	1947	597.1	0.307
[1·PE1/7·PE3/1·PE2]	7.6%	13.2%	79.2%	1910	586.1	0.307
[4·PE2/1·PE1/3·PE3]	8.0%	56.1%	33.3%	1805	542.9	0.301
[6·PE3/1·PE1/2·PE2]	7.4%	26.0%	66.6%	1947	581.7	0.299
[1·PE2/1·PE1/7·PE3]	7.6%	13.2%	79.2%	1910	561.6	0.294
[2·PE2/6·PE3/1·PE1]	7.4%	26.0%	66.6%	1947	563.2	0.289
[1·PE1/3·PE3/4·PE2]	8.0%	56.1%	33.3%	1805	520.0	0.288

## Data Availability

Not applicable.

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
