# Peer review of "Use of Artificial Neural Networks to Optimize Stacking Sequence in UHMWPE Protections"

_polymers, 2021, doi:10.3390/polym13071012_

Round 1

Reviewer 1 Report

(1) These two words, “ANN” and “neural networks” in Keywords, are the same. It is recommended to delete one of them. (2) In the conclusions part, “The maximum validation error of the ANN was 0.89%.”, in my opinion, the maximum validation error of ANN is not convincing. The verification error of 20 runs of ANN may be more convincing. (3) The ordinate of the figures 9-12 can be adjusted to avoid large blanks in these figures, and to make the relationship between the points in the figures more obvious, especially in figure 12.

Reviewer 2 Report

Article Review—March, 2021

Use of artificial neural networks to optimize stacking sequence 2 in UHMWPE protections

 Jairo Peinado, Liu Jiao-Wang, Álvaro Olmedo and Carlos Santiuste

 The aim of the present work is to provide a methodology to evaluate the influence of stacking sequence on the ballistic performance of UHMWPE protections, combining  experimental tests, numerical modelling, and Artificial Neural Networks (ANN). They used ANN modeling to find the best

 stacking sequence when combining layers of three UHMWPE materials with different qualities, is better than one highest performance, leading to cost reduction with a better ballistic limit performance of the UHMWPE protections. Yet, the ANN modeling showed that happens when the weight ratio of the highest-performance material is 79.2%.

Evaluation:

Abstract—Goes into details and repetitions, revise lines 22-to-26 looking at lines 18-to-21

                    The terms UHMWPE (Ultra-High Molecular Weight Polyethylene) & FEM (Finite Element Method ) are NOT defined or spelled out, at
                     all in the paper!!!

It is very confusing to use 1, 2, … to label section, they should have sued I, II, III, IV … instead!!!

  1. Introduction—I am not an expert in the field, I am reviewing the ANN modeling, yet the introlooks well
    written to me.

  1. MethodologyTheir indication that the dataset used to train the ANN is not direct from actual excremental
    data, instead it is obtained via a validated FEM using the experimental data, that worries me
    a bit about accumulated approximations and the credibility of their ANN modeling!  

   2.1 Numerical model—I did not find a clear connection between the NM and the ANN with respect to the
                                            training ANN dataset! In addition, the authors indicated that they did not include in
                                            the ANN training dataset of NM while modeling composites under dynamic loads,
                                            thermal softening and the influence of strain rate to reduce computational cost
                                             because numerous configurations were needed to train the ANN. This will also lower
                                             the ANN modeling credibility, and they should write a note about that!

  2.3 ANN—  A very nonscientific and naïve definition (not in a journal paper!): “An ANN is a connected group 204
                                     of artificial neurons that uses a computational algorithm model to give an answer after 205 processing and interpreting
                              information.”

                              Please get another definition that capture the true mechanism of ANN, and justify its use in your modeling if you
                              already have other means! Lines 204 to 211 need to be revised to a better technical term, than this vague low level
                              style. Although, Bp training has tons of parameters to be adjusted, the authors focused on some only, which served
                              their purpose, yet no mentioning about other parameters. Examples of such parameter, Random initialization of
                              weights which affects the reaching of global minima, Nguyen-Widrow approach for weight initialization as a function
                              of input/hidden layers number of neurons and scale factor b, the format and number of the ANN training patterns,
                             also the time to train the ANN. Other important variations of training Bp ANN are not tried, such as Bp with
                              momentum, batch updating of weights, adaptive learning weight, Delta-Bar-Delta. There many other types of
                             activation functions in addition to the tangent sigmoid.

                             I did not see what was the input to the MATLAB unit and it is output! What is the format and an example vector of the
                             input dataset, and it associated output form the MATLAB? Show for each experiment, what are the involved system
                             parameters in the training vectors’ fields.

                             Also, was the input dataset rich enough and well balanced to justify obtained results?

                        Fig. 6 has be in the same page as its caption line 396-399.

Conclusion—Revise the sentence in line 548  “Two ANNs are used to conduct two studies” it gives the impressions that they
                              used two different ANN models, which is not the case there is only one utilized ANN model MLP-Bp!

                               Lines 543-571: this is a conclusion where you write only the first para you have then you go ahead and state your
                               observations while doing the project and any difficulties you faces, and suggested future work for unfinished
                                business, it looks for me more like a summary!

                                Lines 573-575: I smell contradiction, or at least unsubstantiated very big claim, please rewrite in a lower tone!

Self-Citation:   Olmedo  [13] [33],  Santiuste [1, 11, 12, 13, 33, 34]
